# Bacteriologically confirmed extrapulmonary tuberculosis and the associated risk factors among extrapulmonary tuberculosis suspected patients in Ethiopia: A systematic review and meta-analysis

**Getu Diriba**[1]*, **Ayinalem Alemu**[1,2], **Kirubel Eshetu**[3], **Bazezew Yenew**[1], **Dinka Fikadu Gamtesa**[1], **Habteyes Hailu Tola**[1]

**1** Ethiopian Public Health Institute, Addis Ababa, Ethiopia, **2** Aklilu Lemma Institute of Pathobiology, Addis Ababa University, Addis Ababa, Ethiopia, **3** USAID Eliminate TB Project, Management Sciences for Health, Addis Ababa, Ethiopia

* getud2020@gmail.com

## Abstract

### Background

The actual burden of bacteriologically confirmed extrapulmonary tuberculosis (EPTB) and risk factors in Ethiopia is not well known due to the lack of a strong surveillance system in Ethiopia. Thus, this study was conducted to estimate the pooled prevalence of bacteriologically confirmed EPTB and the associated risk factors among persons suspected to have non-respiratory tuberculosis in Ethiopia.

### Methods

A systematic review and meta-analysis of published studies reporting the prevalence of EPTB from searched electronic databases; Science Direct, PubMed, and Google Scholar was estimated spread across the research periods, nationally, and in different areas, using a fixed-effects model. We used $I^2$ to analyze heterogeneity in the reported prevalence of bacteriologically confirmed extrapulmonary tuberculosis.

### Results

After reviewing 938 research articles, 20 studies (19 cross-sectional and 1 retrospective) from 2003 to 2021 were included in the final analyses. The pooled prevalence of bacteriologically confirmed EPTB was 43% (95%CI; 0.34–0.52, $I^2$ = 98.45%). The asymmetry of the funnel plot revealed the presence of publication bias. Specifically the pooled prevalence of bacteriologically confirmed EPTB based on smear microscopy, Xpert MTB/RIF assay, and culture were 22% (95%CI; 0.13–0.30, $I^2$ = 98.56%), 39% (95%CI; 0.23–0.54, $I^2$ = 98.73%) and 49% (95%CI; 0.41–0.57, $I^2$ = 96.43%) respectively. In this study, a history of pulmonary tuberculosis (PTB) contact with PTB patients, contact with live animals, consumption of raw

**Data Availability Statement:** All the important information is available within the manuscript and its supplementary files.

**Funding:** The author(s) received no specific funding for this work.

**Competing interests:** The authors have declared that no competing interests exist.

milk, HIV-positive, male, and lower monthly income, were found to be independently associated with bacteriologically confirmed EPTB.

## Conclusion

Ethiopia has a high rate of bacteriologically confirmed EPTB. A history of previous PTB, being HIV-positive and having contact with PTB patients were the most reported risk factors for EPTB in the majority of studies. Strengthening laboratory services for EPTB diagnosis should be given priority to diagnose EPTB cases as early as possible.

## Introduction

Pulmonary tuberculosis (PTB), which affects the lungs, accounts for 85% of all reported tuberculosis cases globally [1]. However, extrapulmonary tuberculosis (EPTB), which affects parts of the body other than the lung is becoming a major concern in TB prevention and control efforts [2]. The prevalence of EPTB among notified TB cases in the African region in the 2018 global report was 16%, which is the second-highest next to the Eastern Mediterranean region (24%), which is more than the global prevalence of EPTB (15%). The lowest prevalence was reported in the Western Pacific region (8%) [3]. EPTB is assumed to be produced by the spread of bacteria through the bloodstream from a primary focus in the lung, and hence represents a disseminated form of tuberculosis. EPTB most commonly affects the lymph nodes, abdomen, pleura, bones, and meninges, but the prevalence varies with age, sex, and geographic location [4].

Extrapulmonary TB remains a critical concern both in developing and developed countries. EPTB accounts for 15% to 30% of all tuberculosis cases [5,6]. In persons with HIV/AIDS and other immunocompromised states, it is a prevalent opportunistic infection [7]. In Ethiopia, there is a scarcity of evidence on bacteriological diagnosis and evaluation of EPTB.

Extrapulmonary tuberculosis is difficult to diagnose for a variety of reasons. Many types of EPTB necessitate invasive diagnostic sampling, which can be dangerous to the patient and expensive. Because most types of EPTB are paucibacillary (TB disease caused by a limited number of bacteria), detection by smear microscopy is less sensitive. This particularly affects resource-limited settings, where the more sensitive methods of mycobacterial culture examination are not widely available. Culture has its own set of drawbacks such as a very long turnaround time and necessitating a well-equipped biosafety laboratory [8]. Molecular methods are a quick and sensitive procedure that only requires a small amount of sample and may be used on killed bacteria; however, they require highly trained technologists and can be expensive [9]. As a result of these challenges, EPTB is frequently diagnosed solely based on clinical suspicion, and many people are given the incorrect diagnosis, resulting in needless TB treatment or poor outcomes from untreated EPTB. Even in tertiary health care facilities, the majority of patients had started anti-tuberculosis therapy without bacteriological evidence. These people were misdiagnosed or received therapy too late, and they overestimated the scale of the problem at the community level [10]. In Ethiopia, there are few data on bacteriologically confirmed EPTB among suspected EPTB cases. There is no comprehensive review and meta-analysis of the current research to determine the prevalence of bacteriologically confirmed EPTB among EPTB suspects and its risk factors are poorly understood. s. As a result, this study aimed to investigate the prevalence of bacteriologically confirmed EPTB and the associated risk factors amongst persons suspected to have non-respiratory TB in Ethiopia.

## Materials and methods

### Search strategy

We systematically searched electronic databases such as MEDLINE (PubMed), Science Direct, and grey literature sources such as Google Scholar and Google for articles published in the English language. We used key terms such as "Tuberculosis lymph node", "Tuberculosis cardiovascular", "Tuberculosis central nervous system", "Tuberculosis cutaneous", "Tuberculosis endocrine", "Tuberculosis gastrointestinal", "Tuberculosis hepatic", "Tuberculosis ocular", "Tuberculosis osteoarticular", "Tuberculosis pleura", "Tuberculosis splenic", "Tuberculosis urogenital", and "Ethiopia" both in MeSH and free text.

### Eligibility criteria

We included studies that reported the prevalence of EPTB in Ethiopia. Observational studies, such as cohort (prospective or retrospective) and cross-sectional studies, were included. Studies that were written in English and published before October 26, 2021 (the last date of the searching date), were considered. The studies were included regardless of the diagnostic methods used. The articles without a journal name and/or authors, conference proceedings or presentations, and reviews were excluded from the final analysis.

### Data extraction

We created a data extraction sheet using a Microsoft Excel® 2010 worksheet. Two independent authors (GD, AA) extracted data including study period, study setting (community or facility-based), study site, test method, sample size, and the number of positive patients. The third author (DF) resolved the inconsistencies that arose between the two authors. To ensure consistency, another co-author (HHT) independently examined the extracted data.

### Operational definition

In this meta-analysis, the WHO definition of a positive test result was applied. This states that a positive diagnostic test result using smear microscopy, culture, and Xpert MTM/RIF tests are bacteriological confirmation of EPTB [11].

### Risk of bias assessment and study quality

Two authors (GD and DF) independently assessed the quality of the studies using the Newcastle-Ottawa quality assessment scale adapted for cross-sectional studies [12]. The tool has three components: selection, comparability, and outcome/exposure. The selection part is scored from zero to five stars, and the comparability is scored from zero to two stars. The outcome is scored from zero to three stars. To minimize the subjective interpretation of bias from scoring two reviewers (GD and DF) assessed the quality of individual studies. Furthermore, when the disagreements that occurred throughout the quality grading process were settled by consulting a third author (AA) [12]. We used $I^2$ to analyze heterogeneity in reported prevalence [13,14]. A funnel plot was also used to investigate the presence of publication bias. The presence of publication bias was determined using a funnel plot. By showing funnel plots with the logarithms of effect size and their standard errors, we were able to quantify publication bias.

### Statistical analysis

For statistical analysis, STATA® 14 Stata Corp LLC, Texas, USA software was employed. We estimated the pooled prevalence of bacteriologically confirmed EPTB and a 95% confidence

interval using a fixed-effect meta-analysis model. The 'metaprop' command in STATA 14 was used to determine the pooled prevalence of bacteriologically confirmed EPTB in all patients with EPTB. The distributional information of EPTB was displayed using a forest plot. The pooled effect estimate on the prevalence of bacteriologically confirmed EPTB cases was based on a subgroup analysis of publications comparing culture, smear microscopy, and Xpert MTB/ RIF assay methods.

## Ethical consideration and consent

Since this study is based on previously published articles ethical approval is not applicable.

## Result

### Study selection

The three electronic databases yielded a total of 938 articles, which were then imported into an Endnote X8 citation manager, and 173 duplicates were removed. Next, 722 articles were screened by title and abstract. Around 43 articles that passed the first stage were assessed through a full-text review. During this review, the study subjects, study design, study quality, and outcome were considered. Because of this reason 23 articles were removed. Finally, 20 articles became eligible for data extraction The recommended reporting items for systematic reviews and meta-analysis (PRISMA) flow diagram were used to complete the overall screening (Fig 1).

### Characteristics of the studies included in the review

Of the total of 20 articles reviewed, eight studies were undertaken in the Amhara region [15–22], four studies in Addis Ababa [23–26], four in Oromia [27–30], one study in the Southern Nations Nationalities and People Region [31], and the remaining three studies were based on data collected from different regions of the country [32–34]. The data collection period ranges from 1998 [31] to 2020 [26]. The sample size investigated ranged from 90 [18] to 1,198 participants [34]. The majority of the studies were cross-sectional studies, with one retrospective study. Fourteen studies only examined one type of sample, while the remaining six studies reported evaluating various sample types. Regarding diagnostic methods, five studies used smear microscopy and culture methods [26,29,32–34], three studies used smear microscopy, culture, and Xpert MTB/RIF assay [15,25,28], two studies applied smear microscopy [16,31], two studies used the Xpert MTB/RIF assay [20,22], two studies used the culture and Xpert MTB/RIF assay [19,30], one study used the Xpert MTB/RIF and microscopy [21], and the remaining five studies used the culture [17,18,23,24,27] (Table 1).

### The pooled prevalence of bacteriologically confirmed EPTB

The frequency of EPTB varied widely over the 20 studies. The prevalence ranged from 9% [21] to 78% [32]. The pooled prevalence of bacteriologically confirmed EPTB was 43% (95%CI; 0.34–0.52, $I^2$; 98.45%) according to the random-effects methodology. The highest EPTB prevalence was reported from Addis Ababa, Bar Dar, and Dire Dawa [32], with a rate of 78%, while the lowest was reported from Dessie [21], with a rate of 9% (Fig 2). The pooled proportion of bacteriologically confirmed EPTB studies is represented by a funnel plot (Fig 3). The graph depicted studies with fewer participants and events scattered throughout the pooled horizontal estimate, implying a greater influence due to chance.

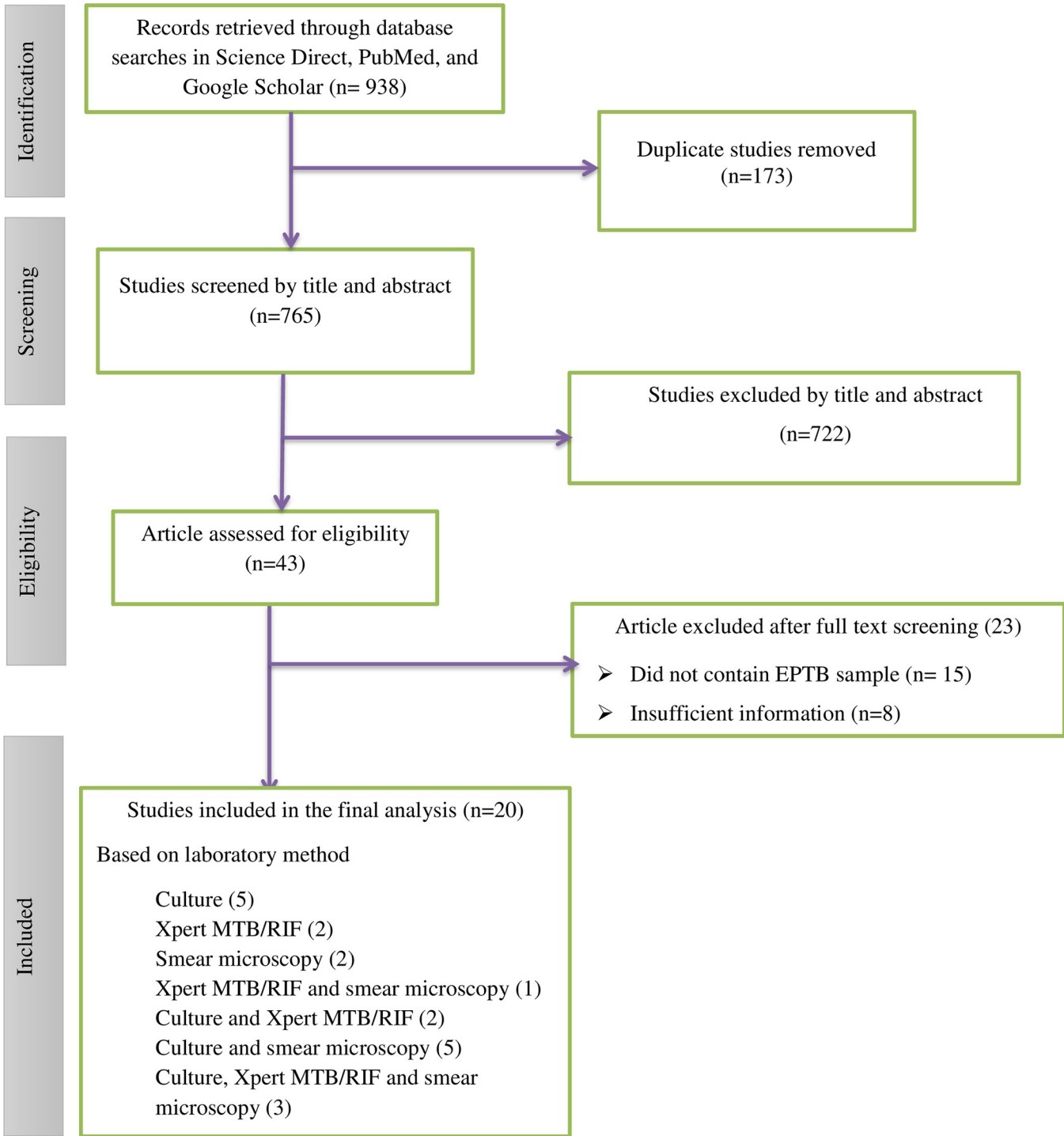

**Fig 1. A PRISMA flow diagram depicting the screening and selection process to identify literature describing extra-pulmonary TB cases in Ethiopia.**

## Subgroup analysis by diagnostic testing methods

There was no heterogeneity among studies conducted in culture, smear microscopy, and Xpert MTB/RIF assay, according to the subgroup analysis diagnostic test. There is heterogeneity among studies that look at multiple diagnostic tests. For culture, smear microscopy, and

**Table 1. General characteristics of studies describing extrapulmonary tuberculosis in Ethiopia.**

| First Author [ref.] | Publication Year | Study Design | Study area | Study Setting | Study time | Sample size (N) | Number of cases with bacteriological confirmation (n/N, %) | Type of EPTB | Diagnostic Method | Risk factors for bacteriologically confirmed TB |
|---|---|---|---|---|---|---|---|---|---|---|
| Yassin et al [31] | 2003 | cross-sectional | Butajira | Facility-based | 1998–2000 | 147 | 107 (72.8) | Lymph node | Microscopy | not stated |
| Iwnetu et al [32] | 2009 | cross-sectional | Addis Ababa, Bar Dar, Diredawa | Facility-based | 2004–2005 | 150 | 117 (78) | Lymph node | Microscopy Culture | not stated |
| Derese et al [33] | 2012 | Retrospective | Woldia, Butajira, Gonder | Facility-based | 2011 | 134 | 50 (37.3) | Lymph node | Microscopy Culture | not stated |
| Biadglegne et al [15] | 2013 | cross-sectional | Bahirdar, Gondar & Dessie | Facility-based | 2012 | 437 | 226 (51.7) | Lymph node | Microscopy Culture Xpert MTB/RIF | Retreated, Male, Age < 14, Urban |
| Zenebe et al [16] | 2013 | cross-sectional | Gonder | Facility-based | 2012 | 344 | 34 (9.9) | Lymph node | Microscopy | history of PTB, raw milk, monthly less income, TB contact |
| Garedew et al [17] | 2013 | cross-sectional | Debre Birhan | Facility-based | 2010–2011 | 98 | 36 (36.7) | Lymph node | Culture | not stated |
| Abdissa et el [27] | 2014 | cross-sectional | Jimma | Facility-based | 2012 | 200 | 147 (73.5) | Lymph node | Culture | not stated |
| Birhanu et al [18] | 2014 | cross-sectional | Dessie | Facility-based | 2012–2013 | 90 | 32 (35.6) | Lymph node | Culture | not stated |
| Berg et al [34] | 2015 | cross-sectional | Gondar, Woldiya, Ghimbi, Fiche, and Butajira, Jinka and Filtu, AA | Facility-based | 2006–2010 | 1198 | 456 (38.1) | Lymph node | Microscopy Culture | having regular and direct contact with live animals, low education level |
| Tadesse et al [28] | 2015 | cross-sectional | Jimma | Facility-based | | 143 | 88 (61.5) | Lymph node | Microscopy Culture Xpert MTB/RIF | not stated |
| Korma et al [23] | 2015 | cross-sectional | Addis Ababa | Facility-based | 2012 to 2013 | 200 | 116 (58) | pleural, peritoneal and synovial fluids | Culture | not stated |
| Abdissa et al [29] | 2015 | cross-sectional | Jimma | Facility-based | 2013 | 144 | 96 (66.7) | Lymph node | Microscopy Culture | not stated |
| Fanosie et al [19] | 2016 | cross-sectional | Gonder | Facility-based | 2015 | 141 | 37 (26.3) | Peritoneal fluid, CSF, Pleural fluid, lymph node | Culture Xpert MTB/RIF | Adult patients, history of contact with known pulmonary TB, HIV positive |
| Zewdie et al [24] | 2016 | cross-sectional | Addis Ababa, | Facility-based | 2013 | 206 | 74 (35.9) | Lymph node | Culture | not stated |
| Mulu et al [20] | 2017 | cross-sectional | Debre Markos | Facility-based | 2014 to 2015 | 182 | 53 (29.1) | Peritoneal, Pus, lymph node, pleural fluid | Xpert MTB/RIF | Retreated, Male, HIV positive, Age 41–50 |
| Metaferia et al [21] | 2018 | cross-sectional | Dessie | Facility-based | 2017 | 353 | 31 (8.8) | Peritoneal fluid, CSF, Pleural fluid, lymph node | Microscopy Xpert MTB/RIF | history of PTB, contact with PTB patients |

(*Continued*)

**Table 1.** (Continued)

| First Author [ref.] | Publication Year | Study Design | Study area | Study Setting | Study time | Sample size (N) | Number of cases with bacteriological confirmation (n/ N, %) | Type of EPTB | Diagnostic Method | Risk factors for bacteriologically confirmed TB |
|---|---|---|---|---|---|---|---|---|---|---|
| Tadesse et al [30] | 2018 | cross-sectional | Jimma | Facility-based | 2015–2017 | 572 | 242 (42.3) | Lymph node, CSF, pleural, peritoneal, and pericardial fluids. | Culture Xpert MTB/RIF | not stated |
| Fantahun et al [25] | 2019 | cross-sectional | Addis Ababa | Facility-based | 2015–2016 | 152 | 75 (49.3) | Lymph node | Microscopy Culture Xpert MTB/RIF | not stated |
| Tedla et al [22] | 2019 | cross-sectional | Dessie | Facility-based | 2018 | 337 | 92 (27.3) | Peritoneal fluid, CSF, Pleural fluid, lymph node synovial fluid | Xpert MTB/ RIF | HIV-positive, history of PTB |
| Assefa et al [26] | 2021 | cross-sectional | Addis Ababa | Facility-based | 2020 | 211 | 50 (23.7) | Lymph node | Microscopy Culture | not stated |

HIV-human immunodeficiency virus; MTB/RIF-Mycobacterium tuberculosis/ Rifampicin; PTB-pulmonary tuberculosis; TB-Tuberculosis.

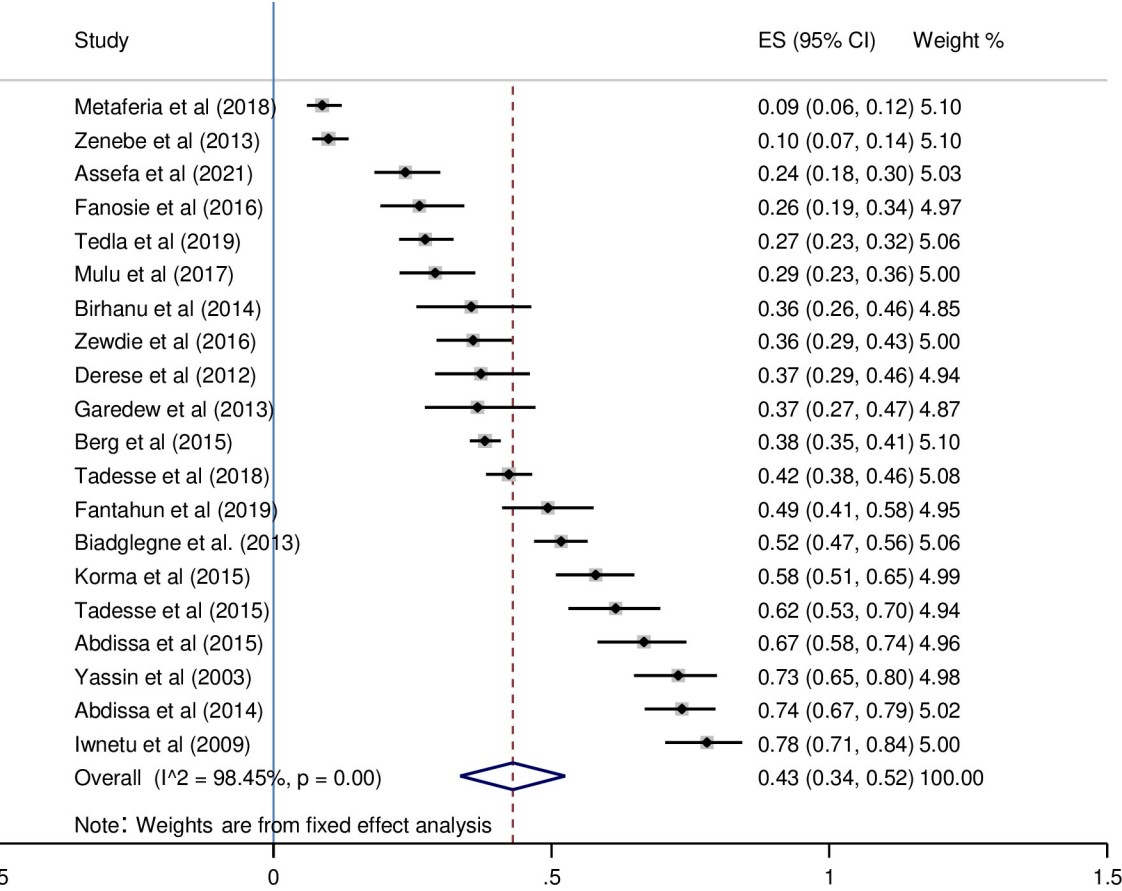

**Fig 2. The pooled proportion of bacteriologically-confirmed EPTB cases amongst all studies identified for review.**

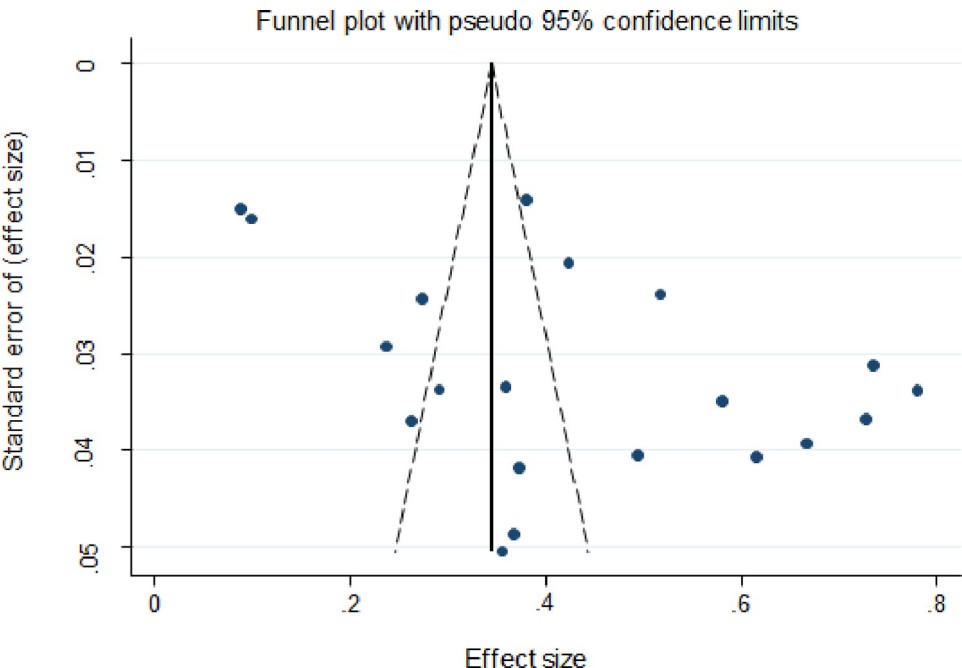

**Fig 3. Funnel plot for the pooled proportion of bacteriologically-confirmed EPTB cases amongst all studies identified for review.**

Xpert MTB/RIF assay diagnostic methods, the prevalence of pooled effect estimates was 49% (95%CI; 0.41–0.57, $I^2$ = 96.43%) (see Fig 4), 22% (95%CI; 0.13–0.30, $I^2$ = 98.56%) (see Fig 5), 39% (95%CI; 0.23–0.54, $I^2$ = 98.73%) (see Fig 6), respectively.

## Risk of bias across studies publication

Visual inspection revealed indications of publication bias for the majority of the culture diagnostic method estimates, with most studies clustered at the funnel's apex and a few spread to the extreme right and left corners (Fig 7). The funnel plots for Xpert MTB/RIF assay and smear microscopy methods were most studies clustered at the funnel's bottom and a few spread to the extreme right corners (Figs 8 and 9).

## Associated risk factors of bacteriologically confirmed EPTB

The impact of each study on the overall meta-analysis summary estimate was investigated. A history of PTB infection and contact with PTB patients was found to be significant risk factors for EPTB incidence in the majority of investigations [15,16,19–22]. Furthermore, having regular and direct contact with live animals, as well as the consumption of raw milk, were found to be strongly related to the incidence of EPTB [16,34]. Additionally, being HIV-positive [19,20,22], ages <14 [15], age 41–50 [20], being male [15,20], monthly less income [16], urban [15] were all linked to the most common EPTB.

## Discussion

This systematic review and meta-analysis estimated the pooled prevalence of bacteriologically confirmed EPTB and the associated risk factors among persons suspected to have non-respiratory tuberculosis in Ethiopia using published studies over the last two decades. This meta-analysis included a total of 5439 EPTB suspects from 20 studies published between 2003 and 2021.

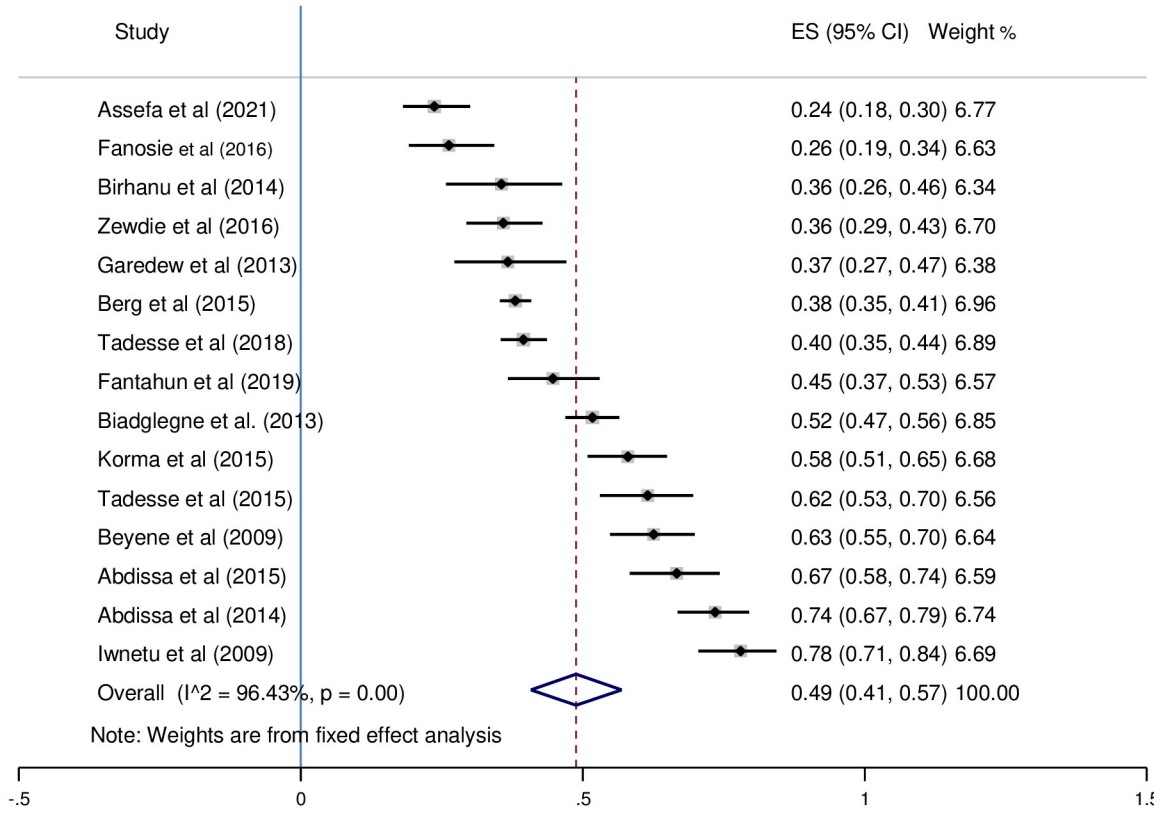

**Fig 4. Pooled proportion of culture-positive EPTB using a fixed-effects model.**

The pooled estimated prevalence of samples with any bacteriological evidence in the studies was 43% (95%CI; 0.34–0.52, $I^2$; 98.45%). A history of PTB infection and contact with PTB patients, contact with live animals, raw milk, HIV, male, less income, and urban, contact with EPTB patients were all found to be independently associated with EPTB in this study.

The overall pooled prevalence of bacteriologically confirmed EPTB in this systematic review and meta-analysis data was 43%. This finding is approximately similar to that previously reported from Cameroon [35]. In contrast, when compared to the estimated prevalence of bacteriologically-confirmed EPTB among all cases of TB in Africa, this is a high figure. According to a 2017 WHO report, the prevalence of EPTB in all cases of TB in Africa and the rest of the globe was 16% and 15%, respectively [7]. Furthermore, a systematic review and meta-analysis of the prevalence of bacteriologically-confirmed EPTB among patients living with HIV/AIDS in Sub-Saharan Africa revealed a lower prevalence of bacteriologically-confirmed EPTB than our findings [36]. However, in the current study, the funnel plot revealed that there is publication bias, where among 20 studies included in this study, 10 were above the 95% upper limit and 5 were below the 95% lower limit and only 5 were within the CI. This might be due to the low number of studies conducted so far in Ethiopia and their variations in

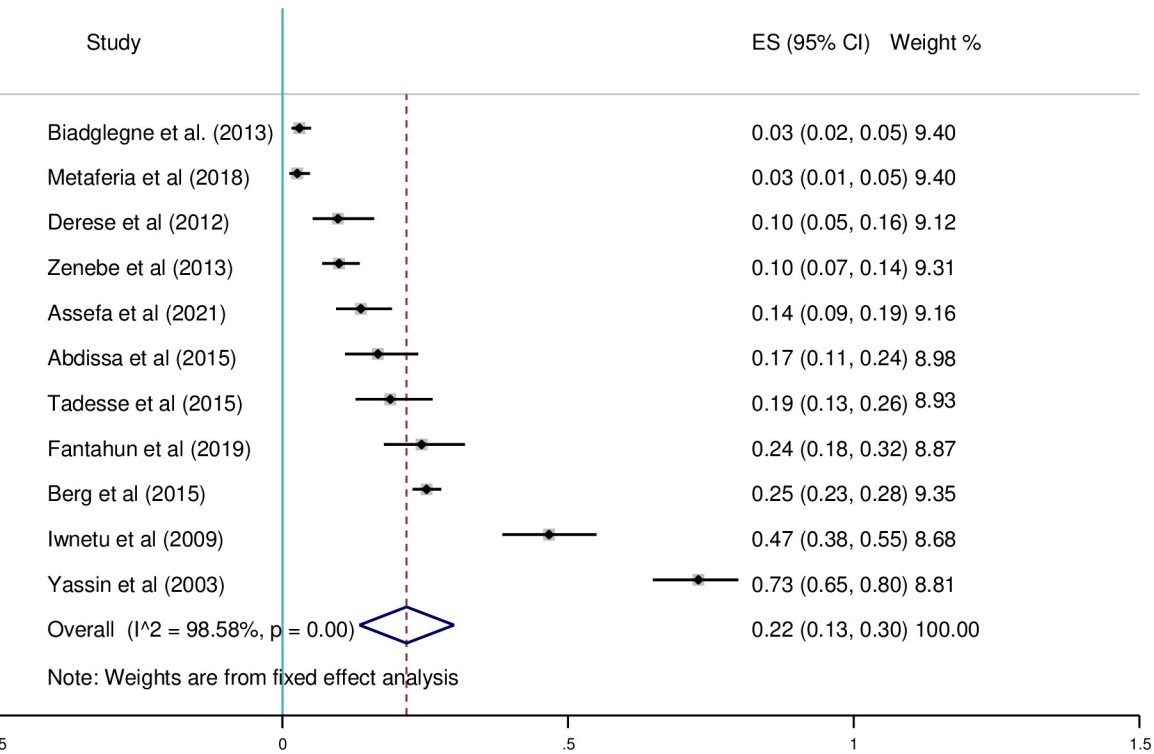

**Fig 5. Pooled proportion of smear-positive EPTB using a fixed-effects model.**

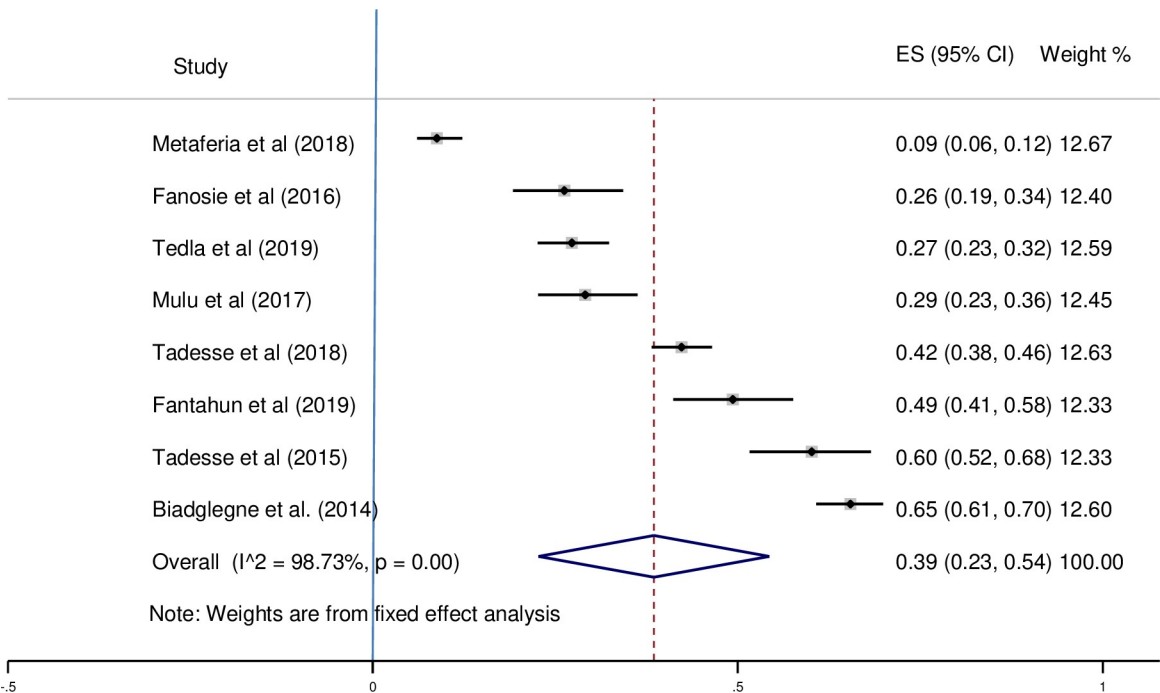

**Fig 6. Pooled proportion of Xpert MTB/RIF assay positive EPTB using a fixed-effects model.**

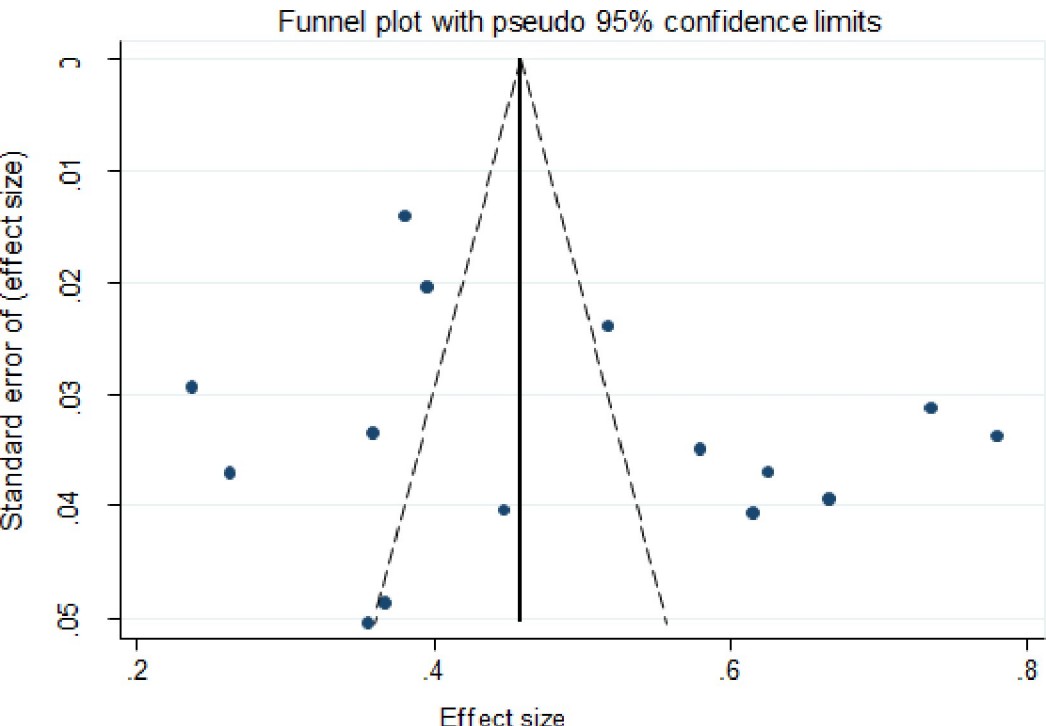

**Fig 7. Funnel plot of a subgroup of 15 of the 20 selected studies for culture-positive EPTB.**

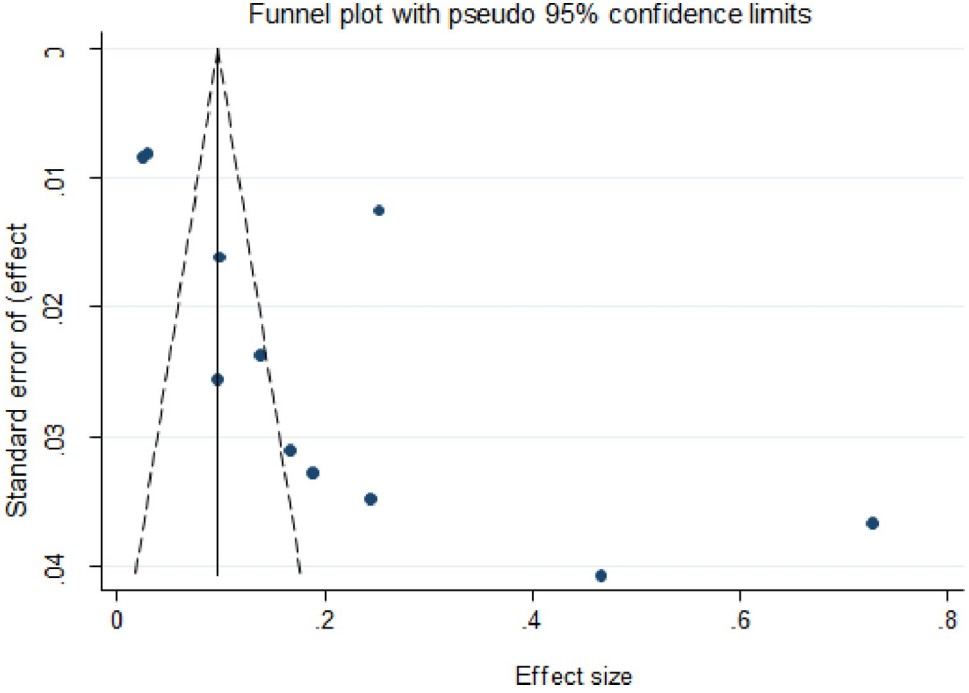

**Fig 8. Funnel plot of a subgroup of 11 of the 20 selected studies for smear-positive EPTB.**

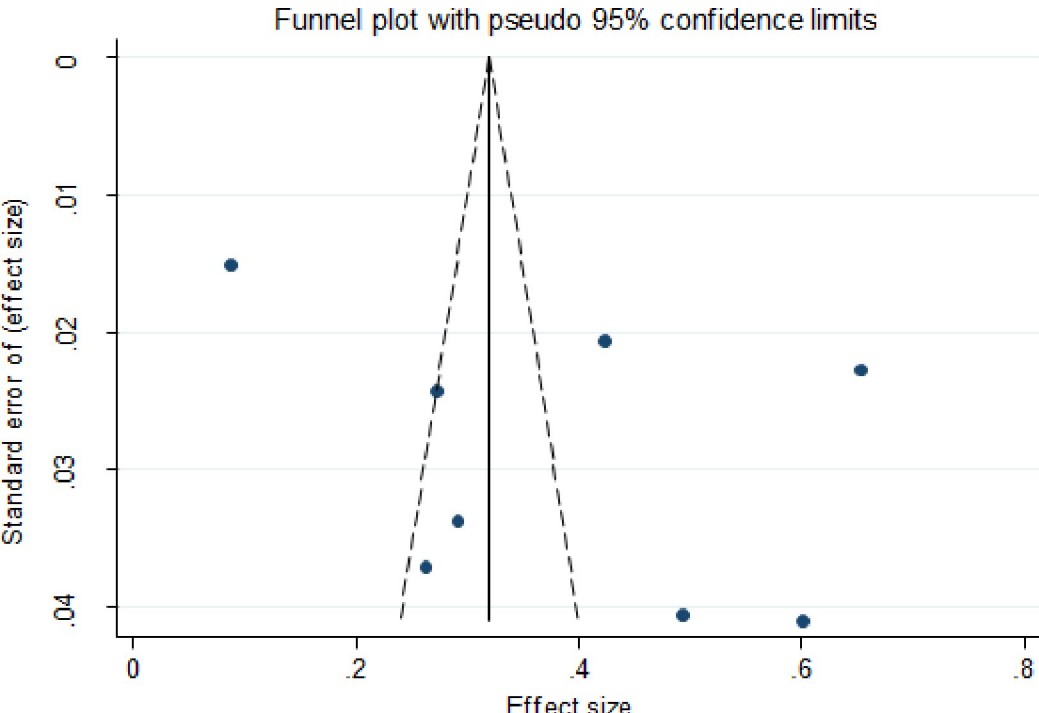

**Fig 9. Funnel plot of a subgroup of 8 of the 20 selected studies for Xpert MTB/RIF assay positive EPTB.**

using laboratory methods. Thus, this may underestimate the pooled bacteriologically confirmed EPTB prevalence among EPTB presumptive cases in Ethiopia.

In this current systematic review and meta-analysis investigation, HIV-1 infected patients were the most significant risk factors for EPTB infection [19,20,22]. Similarly, several studies have examined the association between HIV-1 infection and EPTB infection [6,37]. Furthermore, nearly half of EPTB patients were HIV-1 infected, according to a prior study [38]. This is due to the virus's immune deficiency condition, which allows the bacteria to spread from the primary infection site, the lung, to other parts of the body. During TB-HIV-1 co-infection, there is a lack of granuloma growth and functional disruption of the local immune response within the granuloma [39].

Our study showed that having a history of TB and a history of contact with known pulmonary TB patients was found to be significant risk factors for EPTB development [16,19–22]. Similarly, it is well known that patients with a history of anti-TB treatment cases have an increased risk of EPTB [37].

In this study, we also discovered that men had a higher prevalence of EPTB involvement than women [15,20], which is consistent with similar findings in the previous study [40]. However, another systematic review and meta-analysis study in Africa found that refers to women with lymphadenitis with a higher rate of EPTB than men [41]. Likewise, women had a higher rate of EPTB infections than men [42]. In addition, another study reported women with a higher rate of EPTB than men [43].

In the end, the current study had its limitations. Firstly, the degree of EPTB prevalence in many parts of the country has yet to be addressed, making it impossible to conclude the true burden of EPTB in Ethiopia. Secondly, the observed publication bias that could be due to the differences in the laboratory methods might underestimate the estimated prevalence. Thirdly, Only three databases were searched. This could lead to publication bias. Finally, there is high

heterogeneity among studies that might affect the true estimates. However, the findings are still significant, because the rising rate of EPTB patients in the general population is concerning.

## Conclusions

The finding of this study revealed that there is a high bacteriologically confirmed EPTB among persons suspected to have EPTB in Ethiopia. Patients having a history of previous tuberculosis, a poor income, a history of tuberculosis contact with a known PTB case, being HIV-1 positive, and having contact with PTB patients and a history of underlying diseases was with the most reported risk factors for EPTB. Thus, we recommend strengthening laboratory services for the diagnosis of EPTB in Ethiopia.

## Supporting information

**S1 File. PRISMA checklist.**
(DOC)

**S2 File. Literature search strategy from searched databases.**
(DOCX)

**S3 File. Detailed data of the included studies.**
(XLSX)

**S4 File. Newcastle-Ottawa quality assessment scale for cross-sectional studies.**
(DOCX)

## Acknowledgments

We acknowledge Ethiopian Public Health Institute for the access to article searching. We also acknowledge the authors of the original articles included in this systematic review and meta-analysis study.

## Author Contributions

**Conceptualization:** Getu Diriba, Ayinalem Alemu.

**Data curation:** Getu Diriba, Ayinalem Alemu, Kirubel Eshetu, Bazezew Yenew, Habteyes Hailu Tola.

**Formal analysis:** Getu Diriba, Ayinalem Alemu, Kirubel Eshetu, Bazezew Yenew, Dinka Fikadu Gamtesa.

**Funding acquisition:** Getu Diriba, Ayinalem Alemu.

**Investigation:** Getu Diriba, Bazezew Yenew, Dinka Fikadu Gamtesa, Habteyes Hailu Tola.

**Methodology:** Getu Diriba, Ayinalem Alemu, Kirubel Eshetu, Habteyes Hailu Tola.

**Software:** Ayinalem Alemu, Dinka Fikadu Gamtesa.

**Validation:** Ayinalem Alemu, Kirubel Eshetu, Bazezew Yenew.

**Writing – original draft:** Getu Diriba.

**Writing – review & editing:** Ayinalem Alemu, Kirubel Eshetu, Habteyes Hailu Tola.

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
