## [Decision Letter · Decision Letter 0]

10 Mar 2022

PONE-D-22-00257Bacteriologically confirmed extrapulmonary tuberculosis and the associated risk factors among extrapulmonary tuberculosis suspected patients in Ethiopia: A systematic review and meta-analysisPLOS ONE

Dear Dr. Diriba,

Thank you for submitting your manuscript to PLOS ONE. After careful consideration, we feel that it has merit but does not fully meet PLOS ONE’s publication criteria as it currently stands. Therefore, we invite you to submit a revised version of the manuscript that addresses the points raised during the review process.

Both reviewers felt that the paper had merit. Both did however identify concerns with the methodology. Specifically, there was a concern that the denominator of the study (all PTB cases) was not realistic and undefined and 1 reviewer felt strongly that the review question needed to be tightened. There were also concerns with the construction of the forest plots which reviewer 1 felt should be ordered from low to high. Both reviewers highlighted a number of small stylistic errors which should also be corrected.

We look forward to receiving your revised manuscript.

Kind regards,

Elizabeth S. Mayne, M.D.

Academic Editor

PLOS ONE

Journal Requirements:

5. Please ensure that you refer to Figure 5 in your text as, if accepted, production will need this reference to link the reader to the figure.

Additional Editor Comments:

Both reviewers felt that the paper had merit. Both did however identify concerns with the methodology. Specifically, there was a concern that the denominator of the study (all PTB cases) was not realistic and undefined and 1 reviewer felt strongly that the review question needed to be tightened. There were also concerns with the construction of the forest plots which reviewer 1 felt should be ordered from low to high. Both reviewers highlighted a number of small stylistic errors which should also be corrected.

Reviewers' comments:

Reviewer's Responses to Questions

**Comments to the Author**

1. Is the manuscript technically sound, and do the data support the conclusions?

Reviewer #1: Partly

Reviewer #2: Partly

2. Has the statistical analysis been performed appropriately and rigorously? 

Reviewer #1: No

Reviewer #2: N/A

3. Have the authors made all data underlying the findings in their manuscript fully available?

Reviewer #1: Yes

Reviewer #2: Yes

4. Is the manuscript presented in an intelligible fashion and written in standard English?

Reviewer #1: No

Reviewer #2: Yes

5. Review Comments to the Author

Reviewer #1: Line

22 lack of a strong…

24 insert full stop

spread across
insert full stop
“majority”: what is the percentage?

31 substitute “articles” for “research”

variations
“medium/high risk”: the funnel plot in fig 3 for the 20 papers shows 10 papers above the 95% upper limit and 5

below the 95% lower limit and only 5 within the CI. More discussion is required for your assessment. Is this justified

by the standard error of the effect size on the y-axis?

Delete “having”. PTB must be in full here, that is, “pulmonary tuberculosis”
Delete “being”
Keywords:… “Pulmonary Tuberculosis” for Tuberculosis…
Use “EPTB”; this has already been defined
EPTB
15%
Delete “and”
Delete extra space after “resource- “
turnaround times
“had started” for “have been started”
(11) for [11]
et seq. Remove end full stops in numbering, for example, 2. and 2.1. should be 2 and 2.1
Delete “the”
Delete “language”
“using a Microsoft Excel worksheet” should be “Microsoft Excel® 2016” (or whatever version was used)
“resolved inconsistencies that arose”

99-102 The WHO definition of a positive test result was applied. This states that a positive diagnostic test result using

smear microscopy, culture, Xpert MTM/RIF or nucleic acid amplification test is a bacteriological confirmation of an

EPTB infection.

Delete “included”

105-107 Persistent disagreements indicate a lack of clarity in applying the Newcastle-Ottawa scale. Should the scale be

modified and would different interpretations be applied by researchers on similar studies. The nature of the

disagreements and outcomes of using alternative interpretations should be discussed especially in light of the

small sample of 20 papers

STATA® 14.2 StataCorp LLC, Texas, USA
a 95% confidence interval. Random-effects
The ‘metaprop’ command in STATA was used.

115-116 A forest plot shows distributional information not prevalence

119-127 This requires the editing out of excess detail in light of the PRISMA diagram. The PRISMA diagram is important

and sets out clearly the process in reducing the 938 articles to 20 used in this study

The 36 non-Ethiopian papers should either be included as controls in the forest plots or in separate forest plots. These

could indicate possible biases, inconsistencies or differences in interpretation of data in Ethiopian papers (for example

see disagreements in 105-107 above). This applies to funnel plots as well

Delete “was”
Delete “another”

Page 8 Some hyphens missing in “cross-sectional”. Associated factors column requires some editing (spaces and spacing

of commas)

The frequency of EPFB varied widely over the 20 studies. The prevalence ranged from 9% (21) to 78% (32). See also

general comment below on superscript of paper reference.

random-effects
using a random-effects model

157 “Effect size” should appear in full in the x-axis label

“twenty studies”, delete used.

158-159 Standard error of effect size should appear in full on the y-axis

The subgroups of the 20 studies should be shown in the PRISMA diagram to ensure completeness and intelligibility
Pooled proportion of culture-positive EPTB using random-effects model
Same style as 170
Same style as 170

184 Funnel plot of subgroup of 15 of the 20 selected studies

Standard error of effect size and effect size should appear in the axis labels
Same style as 184
Same style as 184
A history of PTB infection
Delete “of”
HIV should be HIV-1
A history of PTB infection ("history" implies previous history)
… similar to that…
“systemic” should be “systematic”
HIV-1
…risk factors associated with EPTB infection…
HIV-1
… EPTB patients were HIV-1 infected,…
TB-HIV-1 co-infection
… with a history of… (“history” implies previous history)
Firstly,
Secondly,
Statistical power and its reduction has not been discussed in the paper. What is the estimate of the reduction and has

this invalidated the paper? (Does statistical power here refer to I^2, p-value or items outside the 95% pseudo-

confidence gradients in the funnel plots?

238 “poor quality of several studies” – I am not sure this has been sufficiently discussed (if at all)

244 HIV-1

General: number references to papers should preferably be shown as superscripts to avoid with number in the paper. For example, line 49, …region (8%) (3) should be …region (8%)3 (3 as superscript)

References: Journal references should be italicised. Latinisms like et.al. should be italicised

Figure 1: PRISMA diagram should show the subgroups budding off the “Final analysis” box. Reference to this in the main body of the paper will simplify some of the wording in the main body

Figure 2: Forest plot should be sorted by ES and show ranking from lowest to highest ES. Metaferia et. al. would be at the bottom and Iwnetu et.al. at the top

Figure 3: Axis-labels to be described in full

Figure 4: Sort the publications as in figure 2

Figure 5: Sort the publications as in figure 2

Figure 6: Sort the publications as in figure 2

Figures 7, 8 and 9 – same as in figure 3

Reviewer #2: This is a well written article, and findings will be helpful to Ethiopian health care workers, and others working in a LMIC/African context.

However, I have a difficulty with the foundational methodology (meta-analysis) of the study.

o It is not stated what the research questions for the descriptive studies of EPTB which were included in the meta-analysis were.

o Studies which were included could have identified EPTB as a subset of all PTB, or they may have described EPTB alone, from persons with clinical criteria matching only those in whom EPTB is suspected.

o Without specifying which approach the authors of this meta-analysis wanted de novo, the selection criteria results in a wide range of approaches to EPTB diagnosis being selected, which will lead to enormous heterogeneity, and lack of meaningful comparison.

Before conducting a meta-analysis, it is critical to given to formulate the review question clearly. I think the review question ‘to investigate the prevalence of EPTB in Ethiopia’ is too broad. It will be helpful to say ‘Investigate the prevalence of EPTB amongst persons with non-pulmonary TB’. The results are then clearly applicable to persons in whom PTB has been excluded.

As the current aim stands, it implies that that the aim is to identify the burden of TB amongst all TB cases in Ethiopia. This begs the question of a denominator. Assessing the prevalence of EPTB requires that one knows the total burden of TB. How is it possible to assess the prevalence by identifying only studies that described EPTB? These studies would have identified persons who were suspected of EPTB as the starting point. Equally, an ascertainment of risk factors can only be done by comparison with non-EPTB groups.

Without narrowing the research question, it is not meaningful to present or interpret Forest plots.

Other minor comments

A small consideration re methodology, search strategy – are there articles published by Ethiopians in literature that would not be listed in pubmed, Science Direct or on google scholar?

Line 49 Extrapulmonary Tuberculosis – ‘tuberculosis’ should be lower case.

Line 64 Turnaround Time – T should be lower case,

Line 70 Anti-TB medicine – rather ‘anti-tuberculosis therapy’

Line 108 The NewCastle Ottowa quality assessment scale is a checklist for cohort and case control studies. What motivated the use of this scale, over and above the PRISMA guidelines which are sufficient for meta-analyses and systematic reviews.

6. PLOS authors have the option to publish the peer review history of their article (what does this mean?). If published, this will include your full peer review and any attached files.

Reviewer #1: No

Reviewer #2: No

---

## [Author Response · Author response to Decision Letter 0]

5 Apr 2022

Ref: PONE-D-22-00257

Bacteriologically confirmed extrapulmonary tuberculosis and the associated risk factors among extrapulmonary tuberculosis suspected patients in Ethiopia: A systematic review and meta-analysis

Dear Editor,

Thank you for your informative comments and thoughtful comments on our manuscript. We have addressed your and the reviewers' comments one by one. We also appreciate the reviewers for their critical observation and informative comments which radically improved our manuscript quality. We have provided point-by-point responses to the reviewers' comments below. Also, we would like to inform you that we have used a hyperlink to indicate where we made changes in the previous version of the manuscript based on the reviewers’ comments.

Journal Requirements:

Response: Thank you for the comment. The manuscript was prepared according to PLOS ONE's style requirements.

Response: Thank you for the valuable comment. In the revised manuscript we included a sentence about ethical consideration “Since this study is based on previously published articles ethical approval is not applicable”.

Response: Thank you. This work did not receive any funding for this study and we have already stated it in the first submission before the reference section “The author(s) received no specific funding for this work”

Response: Thank you. All the important information is available within the manuscript and its supplementary files.

5. Please ensure that you refer to Figure 5 in your text as, if accepted, production will need this reference to link the reader to the figure.

Response: Thank you for your critical observation and comment; we revised as per the suggestion.

Additional Editor Comments:

Both reviewers felt that the paper had merit. Both did however identify concerns with the methodology. Specifically, there was a concern that the denominator of the study (all PTB cases) was not realistic and undefined and 1 reviewer felt strongly that the review question needed to be tightened. There were also concerns with the construction of the forest plots which reviewer 1 felt should be ordered from low to high. Both reviewers highlighted a number of small stylistic errors which should also be corrected.

Response: Thank you for your suggestions and constructive comments. We have addressed your comments across the manuscript methodology. All the forest plots are arranged from low to high. Also, all technical errors have been corrected.

Review Comments to the Author

Reviewer #1 

22 lack of a strong…

Response: Thank you, we revised it accordingly. 

24 insert full stop

Response: Thank you, we corrected it as per the suggestion. 

spread across

Response: Thank you for your suggestion. We have corrected it.

insert full stop

Response: Thank you for your suggestion. We inserted a full stop in the revised manuscript.

“majority”: what is the percentage?

Response: Thank you for the question. At this stage, we revised the result section of the abstract.

substitute “articles” for “research”

Response: Thank you for your suggestion. We revised it accordingly.

variations

Response: Thank you. We have corrected it.

“medium/high risk”: the funnel plot in fig 3 for the 20 papers shows 10 papers above the 95% upper limit and 5 below the 95% lower limit and only 5 within the CI. More discussion is required for your assessment. Is this justified by the standard error of the effect size on the y-axis?

Response: Thank you for the valuable comment. Now we included a detailed paragraph in the discussion section that described it. 

Delete “having”. PTB must be in full here, that is, “pulmonary tuberculosis”

Response: Thank you for the valuable comment. Now, defined it in full first and then we used the abbreviation. 

Delete “being”

Response: Thank you. We have deleted the comment

Keywords:… “Pulmonary Tuberculosis” for Tuberculosis…

Response: Thank you for the suggestion. We revised it. 

Use “EPTB”; this has already been defined

Response: Thank you. In the current version, we used EPTB. 

EPTB

Response: Thank you for your suggestion. We have corrected it. 

15%

Response: Thank you for your comment. We have added the % according to the comment

Delete “and”

Response: Thank you for your suggestion. We have deleted it.

Delete extra space after “resource- “

Response: Thank you. We have deleted the extra space.

turnaround times

Response: Thank you for your suggestion. We corrected it accordingly. 

“had started” for “have been started”

Response: Thank you for your suggestion. We revised it as per the given suggestion. 

(11) for [11]

Response: Thank you for the suggestion. We revised the format for the whole referencing style based on the Plos One journal requirements.

et seq. Remove end full stops in numbering, for example, 2. and 2.1. should be 2 and 2.1

Response: Thank you for your suggestion. In the revised version, we have removed the numbers in the entire manuscript according to the Journal’s criteria.

Delete “the”

Response: Thank you for your suggestion. We deleted it.

Delete “language”

Response: Thank you for your suggestion. We deleted it.

“using a Microsoft Excel worksheet” should be “Microsoft Excel® 2016” (or whatever version was used)

Response: Thank you for your suggestion. We have revised it accordingly.

“resolved inconsistencies that arose”

Response: Thank you for your suggestion. We revised it as per the suggestion

99-102 The WHO definition of a positive test result was applied. This states that a positive diagnostic test result using smear microscopy, culture, Xpert MTM/RIF or nucleic acid amplification test is a bacteriological confirmation of an EPTB infection.

Response: Thank you for the valuable comment and suggestion. We have revised it as per the given suggestion. 

Delete “included”

Response: Thank you, we have deleted it.

105-107 Persistent disagreements indicate a lack of clarity in applying the Newcastle-Ottawa scale. Should the scale be modified and would different interpretations be applied by researchers on similar studies. The nature of the disagreements and outcomes of using alternative interpretations should be discussed especially in light of the small sample of 20 papers

Response: Thank you; we have addressed this according to the comment

STATA® 14.2 StataCorp LLC, Texas, USA

Response: Thank you, we revised it accordingly.

a 95% confidence interval. Random-effects

Response: We have changed according to the comment

The ‘metaprop’ command in STATA was used.

Response: Thank you, we revised it accordingly.

115-116 A forest plot shows distributional information not prevalence

Response: Thank you, it is revised accordingly.

119-127 This requires the editing out of excess detail in light of the PRISMA diagram. The PRISMA diagram is important and sets out clearly the process in reducing the 938 articles to 20 used in this study 123 The 36 non-Ethiopian papers should either be included as controls in the forest plots or in separate forest plots. These could indicate possible biases, inconsistencies or differences in interpretation of data in Ethiopian papers (for example see disagreements in 105-107 above). This applies to funnel plots as well 

Response: Thank you, we revised it as per the suggestion.

Delete “was”

Response: We have deleted it.

Delete “another”

Response: We have deleted it.

Page 8 Some hyphens missing in “cross-sectional”. Associated factors column requires some editing (spaces and spacing of commas)

Response: Thank you, we revised it accordingly.

The frequency of EPFB varied widely over the 20 studies. The prevalence ranged from 9% (21) to 78% (32). See also general comment below on superscript of paper reference.

Response: Thank you for the comment. The numbers in the bracket are the references, now we revised the referencing based on the journal’s criteria or using [ ].

random-effects

Response: We have changed according to the comment

using a random-effects model

Response: We have corrected accordingly.

157 “Effect size” should appear in full in the x-axis label

Response: Thank you. We have addressed the comment

“twenty studies”, delete used.

Response: We have deleted it.

158-159 Standard error of effect size should appear in full on the y-axis

Response: Thank you for the suggestion, we corrected it accordingly. 

The subgroups of the 20 studies should be shown in the PRISMA diagram to ensure completeness and intelligibility

Response: Thank you for the suggestion. We sub-grouped the included studies based on the laboratory method used, and we described it in the flow diagram and early in the result section.

Pooled proportion of culture-positive EPTB using random-effects model

Response: Thank you, we revised it as per the suggestion. 

Same style as 170

Response: We have changed according to the comment

Same style as 170

Response: We have changed according to the comment

184 Funnel plot of subgroup of 15 of the 20 selected studies

Response: We have changed according to the comment

Standard error of effect size and effect size should appear in the axis labels

Response: Thank you for the valuable comment; we revised accordingly per the suggestion. and effect size according to the comment

Same style as 184

Response: Thank you, we revised it accordingly. 

Same style as 184

Response: Thank you, we revised it accordingly.

A history of PTB infection

Response: Thank you, we corrected it. 

Delete “of”

Response: Thank you we have deleted it.

HIV should be HIV-1

Response: Thank you for the suggestion, we revised it accordingly. 

A history of PTB infection ("history" implies previous history)

Response: We have corrected per the suggestion.

… similar to that…

Response: We revised it.

“systemic” should be “systematic”

Response: Thank you, now it is corrected. 

HIV-1

Response: Thank you for the suggestion, we revised it accordingly.

…risk factors associated with EPTB infection…

Response: Thank you, we revised it as per the suggestion. 

HIV-1

Response: Thank you, now it is revised accordingly.

… EPTB patients were HIV-1 infected,…

Response: Thank you. We revised as per the suggestion.

TB-HIV-1 co-infection

Response: Thank you. We revised it accordingly.

… with a history of… (“history” implies previous history)

Response: We have revised it accordingly.

Firstly,

Response: Thank you, we revised it. 

Secondly,

Response: Thank you, we revised it.

Statistical power and its reduction has not been discussed in the paper. What is the estimate of the reduction and has this invalidated the paper? (Does statistical power here refer to I^2, p-value or items outside the 95% pseudo-confidence gradients in the funnel plots? 238 “poor quality of several studies” – I am not sure this has been sufficiently discussed (if at all)

Response: Thank you for the valuable comment. Now, we revised the limitation of the study as follows “In the end, the current study had its limitations. Firstly, the degree of EPTB prevalence in many parts of the country has yet to be addressed, making it impossible to conclude the true burden of EPTB in Ethiopia. Secondly, the observed publication bias that could be due to the differences in the laboratory methods might underestimate the estimated prevalence. Thirdly, there is high heterogeneity among studies that might affect the true estimates. However, the findings are still significant, because the rising rate of EPTB patients in the general population is concerning”

244 HIV-1

Response: We revised it accordingly. 

General: number references to papers should preferably be shown as superscripts to avoid with number in the paper. For example, line 49, …region (8%) (3) should be …region (8%)3 (3 as superscript)

Response: Thank you for the valuable comment and suggestion. Now we revised all the references according to the PloS One guideline, Such that we used the square brackets “[ ]”.

References: Journal references should be italicised. Latinisms like et.al. should be italicized

Response: Thank you, we revised as per the suggestion. 

Figure 1: PRISMA diagram should show the subgroups budding off the “Final analysis” box. Reference to this in the main body of the paper will simplify some of the wording in the main body

Response: Thank you for the suggestion, as we have stated in the previous response we revised the flow diagram according to the PRISMA guideline. 

Figure 2: Forest plot should be sorted by ES and show ranking from lowest to highest ES. Metaferia et. al. would be at the bottom and Iwnetu et.al. at the top

Response: Thank you for the comment, we revised it accordingly. 

Figure 3: Axis-labels to be described in full

Response: We have described in full according to the comment

Figure 4: Sort the publications as in figure 2

Response: Thank you, we revised it accordingly. 

Figure 5: Sort the publications as in figure 2

Response: Thank you, we revised it accordingly. 

Figure 6: Sort the publications as in figure 2

Response: Thank you, we revised it accordingly. 

Figures 7, 8 and 9 – same as in figure 3

Response: Thank you, we revised it accordingly. 

Reviewer #2: 

This is a well written article, and findings will be helpful to Ethiopian health care workers, and others working in a LMIC/African context.

However, I have a difficulty with the foundational methodology (meta-analysis) of the study.

o It is not stated what the research questions for the descriptive studies of EPTB which were included in the meta-analysis were.

o Studies which were included could have identified EPTB as a subset of all PTB, or they may have described EPTB alone, from persons with clinical criteria matching only those in whom EPTB is suspected.

o Without specifying which approach the authors of this meta-analysis wanted de novo, the selection criteria results in a wide range of approaches to EPTB diagnosis being selected, which will lead to enormous heterogeneity, and lack of meaningful comparison.

Before conducting a meta-analysis, it is critical to given to formulate the review question clearly. I think the review question ‘to investigate the prevalence of EPTB in Ethiopia’ is too broad. It will be helpful to say ‘Investigate the prevalence of EPTB amongst persons with non-pulmonary TB’. The results are then clearly applicable to persons in whom PTB has been excluded.

As the current aim stands, it implies that that the aim is to identify the burden of TB amongst all TB cases in Ethiopia. This begs the question of a denominator. Assessing the prevalence of EPTB requires that one knows the total burden of TB. How is it possible to assess the prevalence by identifying only studies that described EPTB? These studies would have identified persons who were suspected of EPTB as the starting point. Equally, an ascertainment of risk factors can only be done by comparison with non-EPTB groups.

Without narrowing the research question, it is not meaningful to present or interpret Forest plots.

Response: Thank you for the pertinent comments and valuable suggestions. This study is designed to investigate/estimate/ the prevalence of bacteriologically confirmed EPTB among individuals who are presumptive to have EPTB. Thus, the denominator is the number of EPTB or non-respiratory TB presumptive individuals, while the numerator is the number of bacteriologically confirmed EPTB presumptive individuals. Now we described in detail the research questions and the objectives in the revised manuscript. We revised the abstract section, the objectives at the end of the introduction section, and the first paragraph in the discussion section. 

Other minor comments

A small consideration re methodology, search strategy – are there articles published by Ethiopians in literature that would not be listed in pubmed, Science Direct or on google scholar?

Response: Thank you very much for your informative comments. The majority of the articles published from Ethiopia are open access, because Ethiopia is among the low-income countries which waived article processing charges. Thus, all open access articles can be accessed through Google Scholar research. In addition, Since we cannot freely access other databases in Ethiopia we could not search them. To address this comment we have added a sentence that indicates as our search is not comprehensive in limitation.

Line 49 Extrapulmonary Tuberculosis – ‘tuberculosis’ should be lower case.

Response: Thank you for the comment, we revised it accordingly. 

Line 64 Turnaround Time – T should be lower case,

Response: Thank you, we revised it as per the suggestion. 

Line 70 Anti-TB medicine – rather ‘anti-tuberculosis therapy’

Response: Thank you, we revised it accordingly. 

Line 108 The NewCastle Ottowa quality assessment scale is a checklist for cohort and case control studies. What motivated the use of this scale, over and above the PRISMA guidelines which are sufficient for meta-analyses and systematic reviews.

Response: Thank you for the valuable question. This systematic review and meta-analysis were conducted following the PRISMA guidelines. We used the Newcastle-Ottawa quality assessment checklist to assess the quality of individuals studies based on the questions available in the checklist as the Joanna Briggs critical appraisal tool. Since all studies included in the current study are cross-sectional studies we have used the Newcastle-Ottawa quality assessment scale adapted for cross-sectional studies. 

Yours, Sincerely 

Getu Diriba

---

## [Decision Letter · Decision Letter 1]

23 May 2022

PONE-D-22-00257R1Bacteriologically confirmed extrapulmonary tuberculosis and the associated risk factors among extrapulmonary tuberculosis suspected patients in Ethiopia: A systematic review and meta-analysisPLOS ONE

Dear Dr. Diriba,

Thank you for submitting your manuscript to PLOS ONE. After careful consideration, we feel that it has merit but does not fully meet PLOS ONE’s publication criteria as it currently stands. Therefore, we invite you to submit a revised version of the manuscript that addresses the points raised during the review process.

Although both readers felt that the manuscript was substantially improved, there were still significant issues with the statistical analysis as highlighted by Reviewer 1. This, given that the rationale for the study was also unclear and the fact that there are concerns regarding incorrect usage of p values, this manuscript still requires extensive editing.

We look forward to receiving your revised manuscript.

Kind regards,

Elizabeth S. Mayne, M.D.

Academic Editor

PLOS ONE

Additional Editor Comments (if provided):

Although both readers felt that the manuscript was substantially improved, there were still significant issues with the statistical analysis as highlighted by Reviewer 1. This, given that the rationale for the study was also unclear and the fact that there are concerns regarding incorrect usage of p values, this manuscript still requires extensive editing.

Reviewers' comments:

Reviewer's Responses to Questions

**Comments to the Author**

1. If the authors have adequately addressed your comments raised in a previous round of review and you feel that this manuscript is now acceptable for publication, you may indicate that here to bypass the “Comments to the Author” section, enter your conflict of interest statement in the “Confidential to Editor” section, and submit your "Accept" recommendation.

Reviewer #1: (No Response)

Reviewer #2: (No Response)

2. Is the manuscript technically sound, and do the data support the conclusions?

Reviewer #1: Partly

Reviewer #2: Partly

3. Has the statistical analysis been performed appropriately and rigorously? 

Reviewer #1: No

Reviewer #2: I Don't Know

4. Have the authors made all data underlying the findings in their manuscript fully available?

Reviewer #1: No

Reviewer #2: Yes

5. Is the manuscript presented in an intelligible fashion and written in standard English?

Reviewer #1: Yes

Reviewer #2: No

6. Review Comments to the Author

Reviewer #1: 1 Reviewer 1 comments in the first review have been addressed and the original document has been extensively revised or rewritten. As a result, the revised document is more readable. The revised document has however highlighted additional significant matters and some minor issues. In view of the novelty and importance of the study, it is well worthwhile to resolve these matters and issues.

Lines refer to the revised document.
Lines 29, 33, 36, 37,117,163,164,180,181,182, and 212. The I^2 test for heterogeneity has values of 98.45%, 98.56%, 98.73% and 96.43%. In terms of Cochran’s Q statistic I^2=((Q-df)/Q*100% (see Cochrane Handbook, 2011), heterogeneity greater than 75% means that studies in the forest plot are not sufficiently comparable and that a meta-analysis may be invalid. A possible remedy is to ignore heterogeneity and adopt a fixed effects model.
Lines 170, 183, 184, 185 and Figures 4, 5, and 6. In terms of point 3 (for a I^2 statistic > 75%) a fixed effects model is usually applied unless a suitable analytical justification is given for the application of a random effects model (see Cochrane Handbook, 2011).
Lines 116 and 117, a rule of thumb for heterogeneity of I^2 of 50% is moderate and I^2 ≥ 75% is high – see point 3 above. The unqualified statement in line 117 of I^2 ≥ 50% would seem to imply that high heterogeneity is desirable.
Lines 180 to 182. P-values of <0.01 are given. P-values as given are incorrectly associated with I^2 tests, which are applied as broad-based categories (see Cochrane Handbook, 2011). P-values are associated with the alternative chi-square test in which a p-value of <0.01 is indicative of a chi-square value in excess of the table value, which indicates high heterogeneity in values and like a high I^2 value (75%), requires further analysis to justify the use of a forest plot (see above).
Repetition of statistics in 5 places - see lines in point 3 above. This suggests that the document requires editing.
Inconsistency in presentation of statistics. The repetitions of statistics in point 7 are inconsistent. Lines 180 to 182, 164 and 212 give p-values, which are not given in the other lines in point 3 above.
The forest plots in Figures 4, 5 and 6 in support of the data in point 3 above have I^2 values >75% which, with p-values <0.01 (presumably relating to a chi-square test), which indicates high heterogeneity. The plots do not indicate if a random effects model or a fixed effects model have been applied. An analysis justifying the relevance of forest plots is required in light of high heterogeneity.
Lines 35 to 37 which refer to smear microscopy, Xpert MTB/RIF, culture as separate categories with statistics for each, does not agree with line 106 which adds nucleic acid amplification as an additional test.
Lines 35 to 37, tests do not agree with the categories in lines 136 to 138, which combine categories. For example, in line 136, three studies combined culture, Xpert MTB/RIF and smear microscopy. The same combination of categories in lines 152 to 157 is at odds with the separate categories in lines 35 to 37. Proposed categories do not agree with tested categories and evaluated categories.
Lines 32, 34, 163, 174, 180-182, 211, 216 do not explain how pooled estimates were computed. Is this a weighted average by subject or by study? How are the 95% confidence intervals computed, are they also weighted by subject or by study?
Line 202 risk factors are live animals, raw milk, HIV, male, less income, urban. Line 215 adds contact with EPTB patients and line 258 adds underlying disease as a risk factor. Line 75 risk of misdiagnosis of tuberculosis is also a risk factor as is line 66 – resource limited settings. Line 243 refers to women with lymphadenitis with a higher rate of EPTB than men and Line 245 as women with a higher rate of EPTB than men contrary to line 202 which regards maleness as being an outright risk.
Lines 256 to 259, in the conclusion, leaves out male, live animals, raw milk, lymphadenitis, and misdiagnosis as risk factors without justification. Consistency in the set of risk factors, from proposal to testing and conclusion should be maintained. Adding or dropping risk factors has not been explained or justified.

Minor points

‘Previous history’ in lines 37, 41, 212, 237, 239, 256, 258 should be ‘history’.

16 /EPTB/ in lines 25 and 80 should be omitted.

Line 45 capitalise f in factors.
Line 99 space between 2010 and worksheet.
Line 120 …software (STATA) so that line 122 refers to ‘STATA’.

Reviewer #2: The authors have clarified the aim of the study, and it is now possible to make comments on the rest of the paper. Please see detailed comments in word document which are summarised here.

Firstly, the rationale for the study could be made clearer.

Secondly, in table 1, it is not clear what 'associated factors' refer to. I think the authors mean 'risk factors', but it is not clear if the factors listed are 'risk factors for EPTB (as opposed to PTB) or 'risk factors for bacteriologically-confirmed TB (vs non-bacteriologically confirmed TB). Following on from this, in the discussion, the authors have made incorrect comparisons with international iterature, and have focused the discussion on prevalence of EPTB amongst all TB cases. Therefore the discussion does not have bearing on the aim of the paper (to discuss % of EPTB that is bacteriologically confirmed, as opposed to the % which is not bacteriologically confirmed). The authors point out that diagnostic tests in Ethiopia are limited, and that understanding the % of EPTB that is bacteriologically confirmed will guide policy. It's not clear in what direction this policy should go. This could be (should be?) brought up in the discussion.

Thirdly, correction of grammatical/syntax errors will make aspects of the paper easier to understand.

Fourthly, all legends to figures and tables should refer to the context of the figure/table in the paper.

Fifthly, once the above are corrected, a statistical reviewer should assess the validity of the tests that have been conducted and how they have been interpreted.

7. PLOS authors have the option to publish the peer review history of their article (what does this mean?). If published, this will include your full peer review and any attached files.

Reviewer #1: **Yes: **Anthony Leland Hamilton Mayne

Reviewer #2: No

---

## [Author Response · Author response to Decision Letter 1]

7 Jun 2022

Ref: PONE-D-22-00257R1

Bacteriologically confirmed extrapulmonary tuberculosis and the associated risk factors among extrapulmonary tuberculosis suspected patients in Ethiopia: A systematic review and meta-analysis

Dear Editor,

Thank you very much for your informative comments on our manuscript entitled “Bacteriologically confirmed extrapulmonary tuberculosis and the associated risk factors among extrapulmonary tuberculosis suspected patients in Ethiopia: A systematic review and meta-analysis”. We have addressed your comments one by one. We also appreciate you for allowing us to revise our manuscript and correct errors in the previous version. We thank the reviewers for their informative comments, and our point-by-point responses to the reviewers’ comments are given below. Also, we would like to inform you that we have used track changes to indicate where we made changes in response to the reviewers’ comments. Review Comments to the Author

Reviewer #1:

 Reviewer 1 comments in the first review have been addressed and the original document has been extensively revised or rewritten. As a result, the revised document is more readable. The revised document has however highlighted additional significant matters and some minor issues. In view of the novelty and importance of the study, it is well worthwhile to resolve these matters and issues.
Lines refer to the revised document.
Lines 29, 33, 36, 37,117,163,164,180,181,182, and 212. The I^2 test for heterogeneity has values of 98.45%, 98.56%, 98.73% and 96.43%. In terms of Cochran’s Q statistic I^2=((Q-df)/Q*100% (see Cochrane Handbook, 2011), heterogeneity greater than 75% means that studies in the forest plot are not sufficiently comparable and that a meta-analysis may be invalid. A possible remedy is to ignore heterogeneity and adopt a fixed effects model.
Lines 170, 183, 184, 185 and Figures 4, 5, and 6. In terms of point 3 (for a I^2 statistic > 75%) a fixed effects model is usually applied unless a suitable analytical justification is given for the application of a random effects model (see Cochrane Handbook, 2011).
Lines 116 and 117, a rule of thumb for heterogeneity of I^2 of 50% is moderate and I^2 ≥ 75% is high – see point 3 above. The unqualified statement in line 117 of I^2 ≥ 50% would seem to imply that high heterogeneity is desirable.
Lines 180 to 182. P-values of <0.01 are given. P-values as given are incorrectly associated with I^2 tests, which are applied as broad-based categories (see Cochrane Handbook, 2011). P-values are associated with the alternative chi-square test in which a p-value of <0.01 is indicative of a chi-square value in excess of the table value, which indicates high heterogeneity in values and like a high I^2 value (75%), requires further analysis to justify the use of a forest plot (see above).

Response: Thank you very much for your unreserved constructive comments. These sections of comments (1-6) have been updated to follow a fixed-effects model, as requested. 

Repetition of statistics in 5 places - see lines in point 3 above. This suggests that the document requires editing.

Response: Thank you for the valuable comment. We have addressed it.

Inconsistency in presentation of statistics. The repetitions of statistics in point 7 are inconsistent. Lines 180 to 182, 164 and 212 give p-values, which are not given in the other lines in point 3 above.

Response: Thank you for the valuable comment. We have corrected it.

The forest plots in Figures 4, 5 and 6 in support of the data in point 3 above have I^2 values >75% which, with p-values <0.01 (presumably relating to a chi-square test), which indicates high heterogeneity. The plots do not indicate if a random effects model or a fixed effects model have been applied. An analysis justifying the relevance of forest plots is required in light of high heterogeneity.

Response: Thank you for the valuable comment. We have used the fixed-effect model and put the type of the model of analysis.

Lines 35 to 37 which refer to smear microscopy, Xpert MTB/RIF, culture as separate categories with statistics for each, does not agree with line 106 which adds

nucleic acid amplification as an additional test.

Response: Thank you for your suggestion. We are removed additional words.

Lines 35 to 37, tests do not agree with the categories in lines 136 to 138, which combine categories. For example, in line 136, three studies combined culture, Xpert MTB/RIF and smear microscopy. The same combination of categories in lines 152 to 157 is at odds with the separate categories in lines 35 to 37. Proposed categories do not agree with tested categories and evaluated categories.

Response: Thank you for your comments. Our findings are reported on lines 35–37 and the characteristics of the research included in the review are described on lines 152–157. We have omitted the repeated lines 136–138.

Lines 32, 34, 163, 174, 180-182, 211, 216 do not explain how pooled estimates were computed. Is this a weighted average by subject or by study? How are the 95% confidence intervals computed, are they also weighted by subject or by study?

Response: Thank you for your critical observations and comments; the pooled estimate was weighted by study.

Line 202 risk factors are live animals, raw milk, HIV, male, less income, urban. Line 215 adds contact with EPTB patients and line 258 adds underlying disease as a risk factor. Line 75 risk of misdiagnosis of tuberculosis is also a risk factor as is line 66 – resource limited settings. Line 243 refers to women with lymphadenitis with a higher rate of EPTB than men and Line 245 as women with a higher rate of EPTB than men contrary to line 202 which regards maleness as being an outright risk.

Response: Thank you for your critical observation and comment; we revised as per the suggestion.

Lines 256 to 259, in the conclusion, leaves out male, live animals, raw milk, lymphadenitis, and misdiagnosis as risk factors without justification. Consistency in the set of risk factors, from proposal to testing and conclusion should be maintained. Adding or dropping risk factors has not been explained or justified

Response: Thank you for your critical observation and comment; we have put the most reported risk factors in the conclusion.

‘Previous history’ in lines 37, 41, 212, 237, 239, 256, 258 should be ‘history’.

Response: Thank you for the suggestion. We revised it.

16 /EPTB/ in lines 25 and 80 should be omitted.

Response: Thank you for your suggestion. We have deleted it.

17 Line 45 capitalise f in factors.

Response: Thank you for your suggestion. We have corrected it. 

18 Line 99 space between 2010 and worksheet.

Response: Thank you for your suggestion. We have corrected it. 

19 Line 120 …software (STATA) so that line 122 refers to ‘STATA’.

Response: Thank you for your suggestion. We corrected it accordingly. 

Reviewer #2: The authors have clarified the aim of the study, and it is now possible to make comments on the rest of the paper. Please see detailed comments in word document which are summarised here.

Firstly, the rationale for the study could be made clearer.

Response: Thank you for your suggestion. We have revised the rational of the study. 

Secondly, in table 1, it is not clear what 'associated factors' refer to. I think the authors mean 'risk factors', but it is not clear if the factors listed are 'risk factors for EPTB (as opposed to PTB) or 'risk factors for bacteriologically-confirmed TB (vs non-bacteriologically confirmed TB). Following on from this, in the discussion, the authors have made incorrect comparisons with international iterature, and have focused the discussion on prevalence of EPTB amongst all TB cases. Therefore the discussion does not have bearing on the aim of the paper (to discuss % of EPTB that is bacteriologically confirmed, as opposed to the % which is not bacteriologically confirmed). The authors point out that diagnostic tests in Ethiopia are limited, and that understanding the % of EPTB that is bacteriologically confirmed will guide policy. It's not clear in what direction this policy should go. This could be (should be?) brought up in the discussion.

Response: Thank you for your suggestion. We have revised it accordingly. 

Thirdly, correction of grammatical/syntax errors will make aspects of the paper easier to understand.

Response: Thank you for your suggestion. In the revised manuscript we have addressed the grammatical problems.

Fourthly, all legends to figures and tables should refer to the context of the figure/table in the paper.

Response: Thank you for your suggestion. We have corrected all the legends accordingly.

Fifthly, once the above are corrected, a statistical reviewer should assess the validity of the tests that have been conducted and how they have been interpreted.

Response: Thank you for your suggestion. Every statistical analysis is correct.

---

## [Decision Letter · Decision Letter 2]

9 Sep 2022

PONE-D-22-00257R2Bacteriologically confirmed extrapulmonary tuberculosis and the associated risk factors among extrapulmonary tuberculosis suspected patients in Ethiopia: A systematic review and meta-analysisPLOS ONE

Dear Dr. Diriba,

Thank you for submitting your manuscript to PLOS ONE. After careful consideration, we feel that it has merit but does not fully meet PLOS ONE’s publication criteria as it currently stands. Therefore, we invite you to submit a revised version of the manuscript that addresses the points raised during the review process.

The reviewers were generally satisfied that the majority of their comments were addressed but there are some minor issues that still need correction. A comprehensive list is included.

We look forward to receiving your revised manuscript.

Kind regards,

Elizabeth S. Mayne, M.D.

Academic Editor

PLOS ONE

Journal Requirements:

Reviewers' comments:

Reviewer's Responses to Questions

**Comments to the Author**

1. If the authors have adequately addressed your comments raised in a previous round of review and you feel that this manuscript is now acceptable for publication, you may indicate that here to bypass the “Comments to the Author” section, enter your conflict of interest statement in the “Confidential to Editor” section, and submit your "Accept" recommendation.

Reviewer #1: (No Response)

2. Is the manuscript technically sound, and do the data support the conclusions?

Reviewer #1: Partly

3. Has the statistical analysis been performed appropriately and rigorously? 

Reviewer #1: Yes

4. Have the authors made all data underlying the findings in their manuscript fully available?

Reviewer #1: Yes

5. Is the manuscript presented in an intelligible fashion and written in standard English?

Reviewer #1: No

6. Review Comments to the Author

Reviewer #1: 1 Please insert spaces or remove spaces as indicated in the red or blue underlines on the document

line 117 "the presence of publication bias..." is a repetition of the previous sentence
line 124 remove STATA 14... details, this has already been defined in line 121
line 171 bacteriologically
line 121 remove (STATA)
line 100 facility-based
line 215 meta-analysis
line 169 Figure 2: pooled not pooed
Table 1 last column is 12 point type, the preceding columns are point 8
The document needs a final edit for typos and set-out

7. PLOS authors have the option to publish the peer review history of their article (what does this mean?). If published, this will include your full peer review and any attached files.

Reviewer #1: **Yes: **Anthony L. H. Mayne

---

## [Author Response · Author response to Decision Letter 2]

10 Sep 2022

Ref: PONE-D-22-00257R2

Bacteriologically confirmed extrapulmonary tuberculosis and the associated risk factors among extrapulmonary tuberculosis suspected patients in Ethiopia: A systematic review and meta-analysis

Dear Editor,

Regarding our manuscript, "Bacteriologically confirmed extrapulmonary tuberculosis and the associated risk factors among extrapulmonary tuberculosis suspected patients in Ethiopia: A systematic review and meta-analysis" we appreciate your insightful comments. Each of your comments has been addressed one by one. We also appreciate your letting us modify our manuscript and make corrections in the first draft. We appreciate the reviewers' informative comments and suggestions. His comments have improved the manuscript effectively. Also, we would like to inform you that we have used hyperlink to indicate where we made changes in the previous version of the manuscript based on the reviewers’ comments.

 Reviewer #1

Please insert spaces or remove spaces as indicated in the red or blue underlines on the document

Response: Thank you for the valuable comment. We have addressed it.

line 117 "the presence of publication bias..." is a repetition of the previous sentence

Response: Thank you for the valuable comment. We have removed the repetition sntence.

line 124 remove STATA 14... details, this has already been defined in line 121

Response: Thank you for your critical observation and comment; we have addressed.

line 171 bacteriologically

Response: Thank you for your suggestion. We have corrected it. 

line 121 remove (STATA)

Response: Thank you for your suggestion. We have remove it. 

line 100 facility-based

Response: Thank you for your suggestion. We have corrected it. 

line 215 meta-analysis

Response: Thank you for your suggestion. We have corrected it. 

line 169 Figure 2: pooled not pooed

Response: Thank you for your suggestion. We have corrected the word. 

Table 1 last column is 12 point type, the preceding columns are point 8

Response: Thank you for the valuable comment. We have addressed in the revised manuscript.

The document needs a final edit for typos and set-out

Response: Thank you for the valuable comment. We have addressed in the revised manuscript.

Kind regards, 

Getu Diriba 

Ethiopian Public Health Institute 

Phone: +251913828019. Fax: +2510112780431

P.O.Box 1242, Addis Ababa, Ethiopia 

E-mail: getud2020@gmail.com

---

## [Decision Letter · Decision Letter 3]

12 Oct 2022

Bacteriologically confirmed extrapulmonary tuberculosis and the associated risk factors among extrapulmonary tuberculosis suspected patients in Ethiopia: A systematic review and meta-analysis

PONE-D-22-00257R3

Dear Dr. Diriba,

We’re pleased to inform you that your manuscript has been judged scientifically suitable for publication and will be formally accepted for publication once it meets all outstanding technical requirements.

Kind regards,

Elizabeth S. Mayne, M.D.

Academic Editor

PLOS ONE

Additional Editor Comments (optional):

Reviewers' comments:

Reviewer's Responses to Questions

**Comments to the Author**

1. If the authors have adequately addressed your comments raised in a previous round of review and you feel that this manuscript is now acceptable for publication, you may indicate that here to bypass the “Comments to the Author” section, enter your conflict of interest statement in the “Confidential to Editor” section, and submit your "Accept" recommendation.

Reviewer #1: (No Response)

2. Is the manuscript technically sound, and do the data support the conclusions?

Reviewer #1: Yes

3. Has the statistical analysis been performed appropriately and rigorously? 

Reviewer #1: Yes

4. Have the authors made all data underlying the findings in their manuscript fully available?

Reviewer #1: Yes

5. Is the manuscript presented in an intelligible fashion and written in standard English?

Reviewer #1: Yes

6. Review Comments to the Author

Reviewer #1: ***please edit spacing and ensure consistent font size is used in tables

Already submitted (review no 2)

7. PLOS authors have the option to publish the peer review history of their article (what does this mean?). If published, this will include your full peer review and any attached files.

Reviewer #1: **Yes: **Anthony Leland Hamilton Mayne

---

## [Editor Report · Acceptance letter]

10 Nov 2022

PONE-D-22-00257R3 

Bacteriologically confirmed extrapulmonary tuberculosis and the associated risk factors among extrapulmonary tuberculosis suspected patients in Ethiopia: A systematic review and meta-analysis 

Dear Dr. Diriba:

I'm pleased to inform you that your manuscript has been deemed suitable for publication in PLOS ONE. Congratulations! Your manuscript is now with our production department. 

Kind regards, 

on behalf of

Dr. Elizabeth S. Mayne 

Academic Editor

PLOS ONE